# Mirror: A Universal Framework for Various Information Extraction Tasks

**Tong Zhu$^\heartsuit$, Junfei Ren$^\heartsuit$, Zijian Yu$^\heartsuit$, Mengsong Wu$^\heartsuit$, Guoliang Zhang$^\heartsuit$, Xiaoye Qu$^\clubsuit$,**
**Wenliang Chen$^{\heartsuit}$*, Zhefeng Wang$^\clubsuit$, Baoxing Huai$^\clubsuit$, Min Zhang$^\heartsuit$**

$^\heartsuit$ Institute of Artificial Intelligence, School of Computer Science and Technology,
Soochow University, China
$^\clubsuit$ Huawei Cloud, China

{tzhu7,jfrenjfren,zjyu,mswumsw,glzhang}@stu.suda.edu.cn
{quxiaoye,wangzhefeng,huaibaoxing}@huawei.com
{wlchen,minzhang}@suda.edu.cn

## Abstract

Sharing knowledge between information extraction tasks has always been a challenge due to the diverse data formats and task variations. Meanwhile, this divergence leads to information waste and increases difficulties in building complex applications in real scenarios. Recent studies often formulate IE tasks as a triplet extraction problem. However, such a paradigm does not support multi-span and n-ary extraction, leading to weak versatility. To this end, we reorganize IE problems into unified multi-slot tuples and propose a universal framework for various IE tasks, namely Mirror. Specifically, we recast existing IE tasks as a multi-span cyclic graph extraction problem and devise a non-autoregressive graph decoding algorithm to extract all spans in a single step. It is worth noting that this graph structure is incredibly versatile, and it supports not only complex IE tasks, but also machine reading comprehension and classification tasks. We manually construct a corpus containing 57 datasets for model pretraining, and conduct experiments on 30 datasets across 8 downstream tasks. The experimental results demonstrate that our model has decent compatibility and outperforms or reaches competitive performance with SOTA systems under few-shot and zero-shot settings. The code, model weights, and pretraining corpus are available at https://github.com/Spico197/Mirror .

## 1 Introduction

Information Extraction (IE) is a fundamental field in Natural Language Processing (NLP), which aims to extract structured information from unstructured text (Grishman, 2019), such as Named Entity Recognition (NER) (Qu et al., 2023b; Gu et al., 2022; Qu et al., 2023a), Relation Extraction (RE) (Cheng et al., 2021), Event Extraction (EE). However, each IE task is usually isolated from specific data structures and delicate models, which makes it

---

*Corresponding author

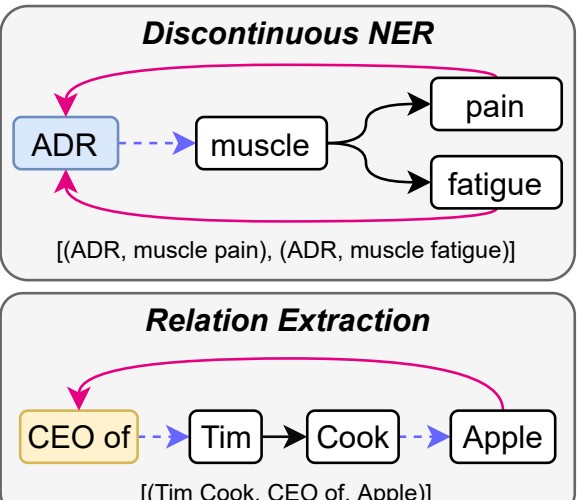

Figure 1: Multi-span cyclic graph for discontinuous NER and RE tasks (best viewed in color). The spans are connected by three types of edges, including ***consecutive connections***, dotted *jump connections* and ***tail-to-head connections***. *ADR* in discontinuous NER refers to the entity label of Adverse Drug Reaction.

difficult to share knowledge across tasks (Lu et al., 2022; Josifoski et al., 2022).

In order to unify the data formats and take advantage of common features between different tasks, there are two main routes in recent studies. The first one is to utilize generative pretrained language models (PLMs) to generate the structured information directly. Lu et al. (2022) and Paolini et al. (2021) structure the IE tasks as a sequence-to-sequence generation problem and use generative models to predict the structured information autoregressively. However, such methods cannot provide the exact positions of the structured information, which is essential to the NER task and fair evaluations (Hao et al., 2023). Besides, the generation-based methods are usually slow and consume huge resources to train on large-scale datasets (Wang et al., 2022). The second way is to apply the extractive PLMs, which are faster to train and inference.

| Model | TANL | UIE | DeepStruct | InstructUIE | USM | Mirror |
|---|---|---|---|---|---|---|
| PLM | T5-base | T5-large | GLM | FlanT5 | RoBERTa | DeBERTa-v3 |
| #Params | 220M | 770M | 10B | 11B | large 372M | large 434M |
| Decoding | AR | AR | AR | AR | NAR | NAR |
| Indexing | Partly | ✗ | ✗ | ✗ | ✓ | ✓ |
| Triplet | ✓ | ✓ | ✓ | ✓ | ✓ | ✓ |
| Single-span NER | ✓ | ✓ | ✓ | ✓ | ✓ | ✓ |
| Multi-span | ✗ | ○ | ○ | ○ | ✗ | ✓ |
| N-ary tuple | ✗ | ✗ | ✗ | ✗ | ✗ | ✓ |
| Classification | ✗ | ✗ | ✗ | ○ | ✗ | ✓ |
| MRC | ✗ | ✗ | ✗ | ✗ | ○ | ✓ |

Table 1: Comparisons among systems. **Circle** ○: the model may support the task theoretically, but the current implementation is not available. **AR**: the auto-regressive decoding while **NAR** is non-autoregressive. **Indexing**: whether the model could provide exact position information. TANL partly supports indexing because the generated tail entity in relation extraction is text-based without position information. Please refer to Appendix A for more detailed comparisons. **Triplet**: "(head, relation, tail)" triplet extraction. **Single-span NER**: flat and nested NER tasks with consecutive spans. **Multi-span**: multi-span extraction, e.g., the discontinuous NER. **N-ary tuple**: the ability of n-ary tuple extraction, e.g., quadruple extraction. **Classification**: the classification tasks. **MRC**: extractive machine reading comprehension tasks. It is worth noting that generative models (TANL, UIE, DeepStruct, and InstructUIE) may be capable of all the tasks if their current paradigms or patterns are changed. However, since the original papers do not contain relevant experiments, we mark them as ✗ or ○ here.

USM (Lou et al., 2023) regards the IE tasks as a triplet prediction problem via semantic matching. However, this method is limited to a small range of triplet-based tasks, and it is unable to address multi-span and n-ary extraction problems.

To overcome the above challenges, we propose *Mirror*, a novel framework that can handle complex multi-span extraction, n-ary extraction, machine reading comprehension (MRC), and even classification tasks, which are not supported by the previous universal IE systems. As exemplified in Figure 1, we formulate IE tasks as a unified multi-slot tuple extraction problem and transform those tuples into multi-span cyclic graphs. This graph structure is rather flexible and scalable. It can be applied to not only complex IE tasks but also MRC and classification tasks. Mirror takes schemas as part of the model inputs, and this benefits few-shot and zero-shot tasks naturally.

Compared with other models in Table 1, Mirror supports efficient non-autoregressive decoding with position indexing and shows good compatibility across different tasks and datasets. We conduct extensive experiments on 30 datasets from 8 tasks, including NER, RE, EE, Aspect-based Sentiment Analysis (ABSA), multi-span discontinuous NER, n-ary hyper RE, MRC, and classification. To enhance the few-shot and zero-shot abilities, we manually collect 57 datasets across 5 tasks into a whole corpus for model pretraining. The experimental results demonstrate that Mirror achieves competitive results under few-shot and zero-shot settings.

Our contributions are summarized as follows:

- We propose a unified schema-guided multi-slot extraction paradigm, which is capable of complex information extraction, machine reading comprehension, and even classification tasks.

- We propose Mirror, a universal non-autoregressive framework that transforms multiple tasks into a multi-span cyclic graph.

- We conduct extensive experiments on 30 datasets from 8 tasks, and the results show that our model achieves competitive results under few-shot and zero-shot settings.

## 2 Related Work

### 2.1 Multi-task Information Extraction

Multi-task IE has been a popular research topic in recent years. The main idea is to use a single model to perform multiple IE tasks. IE tasks could be formulated as different graph structures. Li et al. (2022) formulate flat, nested, and discontinuous NER tasks as a graph with next-neighboring

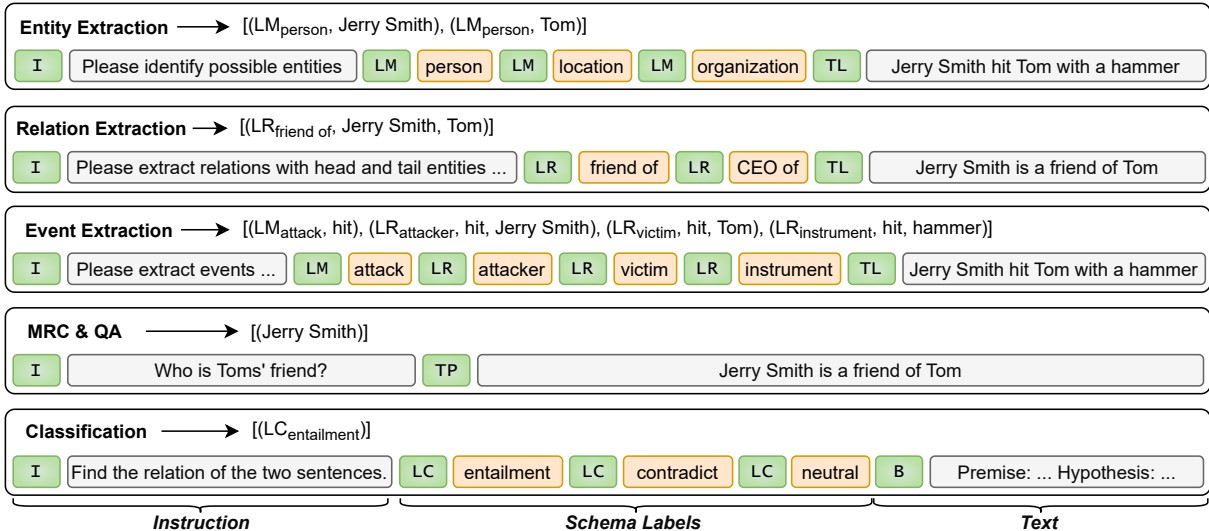

Figure 2: Unified data interface. We design a list of tokens to separate different parts: [I]: instruction. [LM]: mentions. [LR]: relations. [LC]: classifications. [TL]: text that connects with schema labels. [TP]: extractive MRC and QA texts without schema labels. [B]: the background text in the classification task.

and tail-to-head connections. Maximal cliques also have been used to flat & discontinuous NER tasks (Wang et al., 2021) and trigger-available & trigger-free event extractions (Zhu et al., 2022). DyGIE++ takes NER, RE, and EE tasks as span graphs and applies iterative propagation to enhance spans' contextual representations (Wadden et al., 2019). OneIE uses a similar graph structure with global constraint features (Lin et al., 2020).

In addition to explicit graph-based multi-task IE systems, generative language models are widely used. Yan et al. (2021b) and Yan et al. (2021a) add special index tokens into BART (Lewis et al., 2020) vocabulary to help perform various NER and ABSA tasks and obtain explicit span positions. TANL (Paolini et al., 2021) apply T5 (Raffel et al., 2020) to generate texts with special enclosures as the predicted information. GenIE (Josifoski et al., 2022) and DeepStruct (Wang et al., 2022) share a similar idea to generate subject-relation-object triplets, and DeepStruct extends the model size to 10B with GLM (Du et al., 2022).

## 2.2 Schema-guided Information Extraction

In schema-guided IE systems, schemas are input as a guidance signal to help the model extract target information. UIE (Lu et al., 2022) categorize IE tasks into span spotting and associating elementary tasks and devise a linearized query language. Fei et al. (2022) introduces the hyper relation extraction task to represent complex IE tasks like EE, and utilize external parsing tools to enhance

the text representations. InstructUIE (Wang et al., 2023) formulates schemas into instructions and uses FlanT5-11B (Chung et al., 2022) to perform multi-task instruction tuning.

While the above methods utilize generative language models, they cannot predict exact positions, which brings ambiguity when evaluating (Hao et al., 2023). Besides, large generative language models are usually slow to train & infer and require tons of computing resources. USM (Lou et al., 2023) applies BERT-family models to extract triplets non-autoregressively. USM regards IE as a unified schema matching task and uses a label-text matching model to extract triplets. However, these methods cannot extend to complex IE tasks, such as multi-span discontinuous NER and n-ary information extractions.

## 3 Mirror Framework

In this section, we introduce the Mirror framework. We first address the unified data input format to the model, then introduce the unified task formulation and the model structure.

### 3.1 Unified Data Interface

To enable the model to handle different IE tasks, we propose a unified data interface for the model input. As shown in Figure 2, there are three parts: ***instruction***, ***schema labels***, and ***text***. The ***instruction*** is composed of a leading token [I] and a natural language sentence. The [I] token indicates the

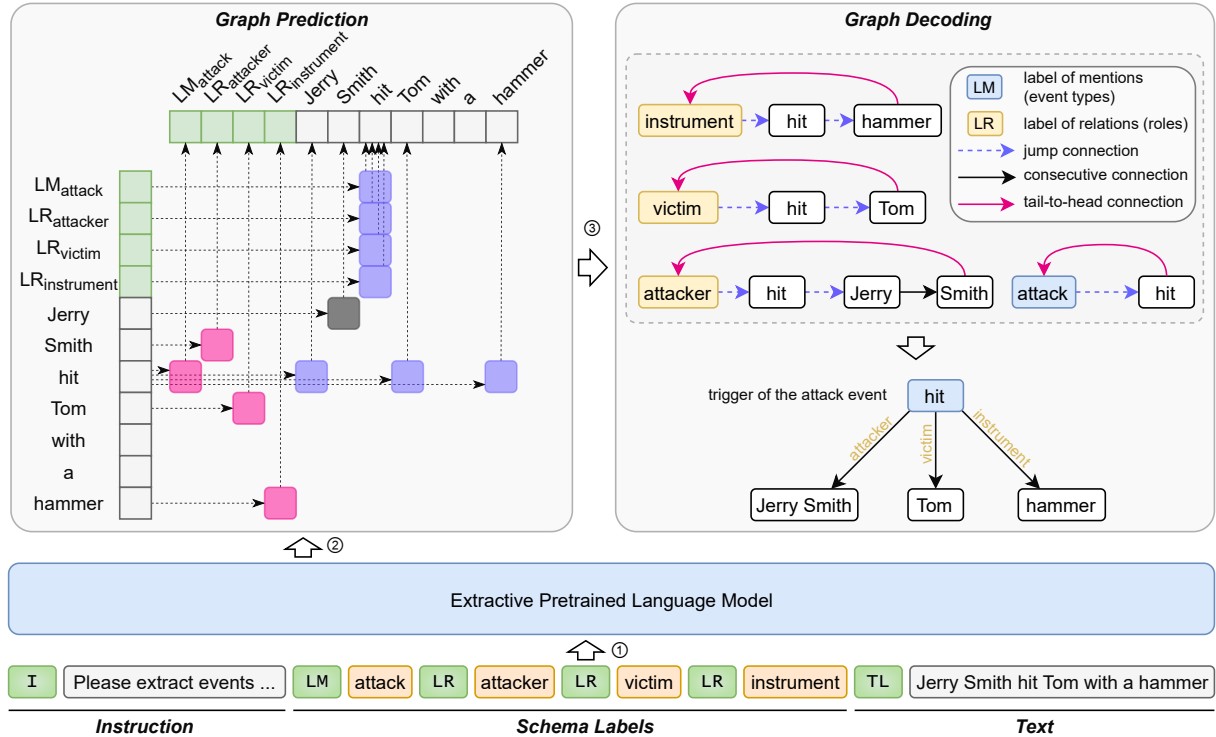

Figure 3: Model framework (best viewed in color). Mirror first constructs inputs for each task, then utilizes a pretrained language model to predict the adjacency matrix via the biaffine attention. After that, final results are decoded from the adjacency matrix accordingly.

instruction part while the sentence tells the model what it should do. For example, the instruction of NER could be "*Please identify possible entities*". In MRC and Question Answering (QA) tasks, the instruction is the question to answer.

The ***schema labels*** are task ontologies for schema-guided extraction. This part consists of special token labels ([LM], [LR], and [LC]) and corresponding label texts. Among the special tokens, [LM] denotes the label of mentions (or event types), [LR] denotes the label of relations (or argument roles), and [LC] denotes the label of classes.

The ***text*** part is the input text that the model should extract information from. It is composed of a leading token ([TL], [TP] or [B]) and a natural language sentence. If the leading token is [TL], the model should link labels from schema labels to spans in the text. While the [TP] token indicates the target spans are only in the text, and the model should extract information from the text without schema labels. In classification tasks, the model should not extract anything from the text part. So we use a special leading token [B] (background) to distinguish it from the extractive text.

With the above three parts, we can formulate extractive MRC, classification, and IE tasks into a

unified data interface, and the model can be trained in a unified way even if the model is not based on generative language models. For the robust model training, we manually collect 57 datasets from 5 tasks to make a corpus for model pretraining. The data statistics for each IE task are listed in Table 2. To balance the number of examples in each task, we set a different maximum number of samples $N_{\max}$ for each task dataset. If the number of instances in a dataset is less than $N_{\max}$, we keep the original dataset unchanged and do not perform oversampling. For NER, RE, and EE tasks, we manually design a set of instructions and randomly pick one of them for each sample. MRC datasets some classification datasets have inborn questions, so the numbers of instruction are much higher than the others. For detailed statistics on each dataset, please refer to Appendix C.

## 3.2 Multi-slot Tuple and Multi-span Cyclic Graph

We formulate IE tasks as a unified multi-slot tuple extraction problem. As exemplified in Figure 2, in the RE task, the model is expected to extract a three-slot tuple: (relation, head entity, tail entity). Here, the tuple is (LR$_{\text{friend of}}$, Jerry

| Task | #Dataset | $N_{\max}$ | #Instruction | #Instance |
|------|----------|-----------|--------------|-----------|
| NER | 15 | 20,000 | 42 | 171,609 |
| Cls♣ | 27 | 5,000 | 54,070 | 134,758 |
| RE | 9 | 20,000 | 9 | 123,876 |
| MRC♡ | 5 | 30,000 | 75,200 | 85,658 |
| EE | 1 | All | 40 | 2,898 |
| Total | 57 | - | - | 518,799 |

Table 2: Pretraining dataset statistics. ♣ Classification tasks contain multi-choice MRC datasets. ♡ MRC stands for both extractive QA and extractive MRC datasets.

Smith, Tom). The length of tuple slots could vary across tasks, so Mirror is able to solve n-ary extraction problems.

As shown in Figure 1 and the top right of Figure 3, we formulate multi-slot tuples into a unified multi-span cyclic graph, and regard labels as the leading tokens in schema labels. There are three types of connections in the graph: the *consecutive* connection, the *jump* connection, and the *tail-to-head* connection. The *consecutive* connection is adopted to **spans in the same entity**. For an entity with multiple tokens, the consecutive connection connects from the first to the last. As shown in Figure 3, "Jerry" connects to "Smith". If there is only one token in an entity, the consecutive connection is not used. The *jump* connection connects **different slots** in a tuple. Schema labels and spans from texts are in different slots, so they are connected in jump connections. For instance, the head and tail entities of a relation triplet are in different slots, so they are connected in jump connections. The *tail-to-head* connection helps **locate the graph boundaries**. It connects from the last token of the last slot to the first token of the first slot in a tuple.

In practice, we convert the answer of each slot into text positions. For schema labels, we use the position of leading tags instead of literal strings. For text spans, the position is a one-digit number if there is only one character, otherwise the start and end positions are listed. For example, the 3-slot relation tuple (LR$_{\text{friend of}}$, Jerry Smith, Tom) will be converted into $(9 \vdots 16 \to 17 \vdots 22)$, where $\vdots$ denotes the jump connection, $\to$ stands for the consecutive connection, 9 is the position of LR$_{\text{friend of}}$, 16 and 17 express *Jerry Smith*, and 22 is the position of *Tom*. There is also a tail-to-head connection from 22 to 9. The corresponding graph decoding algorithm is shown in Algorithm 1. During infer-

---

**Algorithm 1** MULTI-SPAN CYCLIC GRAPH DECODING

**Input:** Adjacency matrix $\mathcal{A}$
**Output:** A set of multi-slot tuples $\mathcal{T}$
1: $\mathcal{T} \leftarrow \{\}$
2: $\tilde{\mathcal{A}} \leftarrow \mathcal{A}^c | \mathcal{A}^j$ ▷ merge consecutive and jump connections
3: Find forward chains $\mathcal{C}$ from $\tilde{\mathcal{A}}$
4: **for** $c \in \mathcal{C}$ **do** ▷ find legal paths with tail-to-head connections
5:     **if** $c$ meets the need in $\mathcal{A}^t$ **then**
6:         split $c$ into a tuple $t$ via $\mathcal{A}^j$
7:         $\mathcal{T} \leftarrow \mathcal{T} \cup t$
8:     **end if**
9: **end for**
10: **return** $\mathcal{T}$

---

ence, we first find the forward chain (9,16,17,22) and then verify the chain with the tail-to-head connection (22→9). After that, the multi-slot tuple is obtained with jump connections$(9 \vdots 16)$ and $(17 \vdots 22)$.

### 3.3 Model Structure

With the unified data interface and the multi-span cyclic graph, we propose a unified model structure for IE tasks. For each token $x_i$ from the inputs, Mirror transforms it into a vector $h_i \in \mathbb{R}^{d_h}$ via a BERT-style extractive pretrained language model (PLM). We use biaffine attention (Dozat and Manning, 2017) to obtain the adjacency matrix $\mathcal{A}$ of the multi-span cyclic graph. Mirror calculates the linking probability $p_{ij}^k, k \in \{\text{consecutive, jump, tail-to-head}\}$ between $x_i$ and $x_j$ as Equation 1 shows. The final $\mathcal{A}$ is obtained via thresholding ($\mathcal{A}_{ij}^k = 1$ if $p_{ij}^k > 0.5$ else 0).

$$\tilde{h}_i = \text{FFNN}_s(h_i), \quad \tilde{h}_j = \text{FFNN}_e(h_j)$$
$$p_{ij}^k = \text{sigmoid}\left(\tilde{h}_i^\top U \tilde{h}_j / \sqrt{d_h}\right), \quad (1)$$

where $\tilde{h}_i, \tilde{h}_j \in \mathbb{R}^{d_b}$. $U \in \mathbb{R}^{d_b \times 3 \times d_b}$ is the trainable parameter, and 3 denotes consecutive, jump, and tail-to-head connections. FFNN is the feedforward neural network with rotary positional embedding as introduced in Su et al. (2021). The FFNN comprises a linear transformation, a GELU activation function (Hendrycks and Gimpel, 2023), and dropout (Srivastava et al., 2014).

During training, we adopt the imbalance-class multi-label categorical cross entropy (Su et al., 2022) as the loss function:

$$\mathcal{L}(i,j) = \log\left(1 + \sum_{\Omega_{\text{neg}}} e^{p_{ij}^k}\right) + \log\left(1 + \sum_{\Omega_{\text{pos}}} e^{-p_{ij}^k}\right) \tag{2}$$

where $\Omega_{\text{neg}}$ stands for negative samples ($\mathcal{A}_{ij}^k = 0$), and $\Omega_{\text{pos}}$ denotes positive samples ($\mathcal{A}_{ij}^k = 1$).

## 4 Experiments

### 4.1 Experiment Setup

We utilize DeBERTa-v3-large (He et al., 2021) as the PLM. The biaffine size $d_b$ is 512 with a dropout rate of 0.3. The epoch number of pretraining is 3 with a learning rate of 2e-5. Please refer to Appendix B for detailed hyper-param settings.

Datasets are processed following Lu et al. (2022) (13 IE datasets in Table 3, and 4 datasets in Table 5), Li et al. (2022) (CADEC in Table 4), Chia et al. (2022) (HyperRED in Table 4), Lou et al. (2023) (7 zero-shot NER datasets in Table 6), Rajpurkar et al. (2018) (SQuAD v2.0 in Table 7), and Wang et al. (2019) (7 GLUE datasets in Table 7). Data statistics and metrics are listed in Appendix C.

### 4.2 Baselines

We compare Mirror with generation-based TANL (Paolini et al., 2021), DeepStruct (Wang et al., 2022), UIE (Lu et al., 2022), InstructUIE (Wang et al., 2023), and extraction-based USM (Lou et al., 2023) in triplet-based IE tasks. In the multi-span discontinuous NER task, we compare Mirror with task-specific BART-NER (Yan et al., 2021b) and W2NER (Li et al., 2022). The baseline system in hyper RE is CubeRE (Chia et al., 2022). As to MRC tasks, the baseline models are BERT (Devlin et al., 2019), RoBERTa (Liu et al., 2019), and DeBERTa-v3 (He et al., 2021).

### 4.3 Main Results

Mirror performances on 13 IE benchmarks are presented in Table 3. Compared with other baseline models, Mirror surpasses baseline models on some datasets in NER (ACE04), RE (ACE05, NYT), and EE (CASIE-Trigger) tasks. When compared to extraction-based USM, Mirror achieves competitive results on most of tasks, while lagging in NER (ACE05), RE (CoNLL04), and EE (CASIE-Arg). Compared to generation-based methods, Mirror outperforms TANL across all datasets and surpasses UIE in most datasets. When the model parameter comes to 10B, DeepStruct outperforms

Mirror on CoNLL04 in the RE task, while Mirror reaches very close results or outperforms DeepStruct on the other datasets. InstructUIE (11B) demonstrates similar performance on NER datasets, while achieving high scores in RE (SciERC) and EE (ACE05-Tgg & Arg), surpassing other models by a significant margin. Apart from these datasets, InstructUIE performs about the same as UIE, USM, and Mirror.

We provide ablation studies on Mirror with different pretraining and fine-tuning strategies. Performance degrades if either pretraining or instruction fine-tuning is not performed. Mirror benefits from pretraining when utilizing instructions (w/ Inst.) and increases 0.66% scores on average. However, when instructions are discarded (w/o Inst.), pretraining (w/ PT) does not bring performance gain. Pretraining has been confirmed on UIE and USM to enhance model performances, and it is crucial to enable the zero-shot inference ability. However, based on the results from Table 3, we find that if Mirror is applied in one specific task with sufficient training resources, it may not need to perform the pretraining step (e.g., NYT dataset).

Besides the traditional IE tasks in Table 3, Mirror also supports multi-span discontinuous NER and n-ary hyper relation extraction as shown in Table 4. We provide Mirror (w/ PT, w/ Inst) and Mirror (w/o PT, w/o Inst.) results on CADEC according to their good performances on the IE tasks in Table 3. However, Mirror is less powerful than task-specific SOTA models. On the n-ary hyper relation extraction task, Mirror outperforms the task-specific model CubeRE and achieves new SOTA results. Table 4 indicates Mirror's compatibility with complex multi-span and n-ary extraction problems.

The above facts indicate that Mirror has good compatibility across different IE problems, and we extend the universal IE system to complex multi-span and n-ary extraction tasks, which are not supported by previous universal IE systems.

### 4.4 Few-shot Results

Followed by Lu et al. (2022) and Lou et al. (2023), we analyze Mirror's few-shot ability on NER, RE, EE, and ABSA tasks. As shown in Table 5, Mirror (w/ PT, w/ Inst.) outperforms USM and achieves SOTA results on CoNLL03, ACE05, and 16-res datasets. In the RE task on CoNLL04, the best model is USM, achieving an average score of 50.12, while Mirror is less effective with only 43.16 av-

| Task | Datasets | TANL | UIE | DeepStruct | InstructUIE | USM | Mirror w/ PT w/ Inst. | Mirror w/ PT w/o Inst. | Mirror w/o PT w/ Inst. | Mirror w/o PT w/o Inst. |
|---|---|---|---|---|---|---|---|---|---|---|
| NER | ACE04 | - | 86.89 | - | - | 87.62 | 87.16 | 86.39 | **87.66** | 87.26 |
| | ACE05 | 84.90 | 85.78 | 86.90 | 86.66 | **87.14** | 85.34 | 85.70 | 86.72 | 86.45 |
| | CoNLL03 | 91.70 | 92.99 | 93.00 | 92.94 | **93.16** | 92.73 | 91.93 | 92.11 | 92.97 |
| RE | ACE05 | 63.70 | 66.06 | 66.80 | - | 67.88 | 67.86 | 67.86 | 64.88 | **69.02** |
| | CoNLL04 | 71.40 | 75.00 | 78.30 | 78.48 | **78.84** | 75.22 | 72.96 | 71.19 | 73.58 |
| | NYT | - | 93.54 | 93.30 | 90.47 | 94.07 | 93.85 | **94.25** | 93.95 | 93.31 |
| | SciERC | - | 36.53 | - | 45.15 | 37.36 | 36.89 | 37.12 | 36.66 | **40.50** |
| EE | ACE05-Tgg | 68.40 | 73.36 | 69.80 | **77.13** | 72.41 | 74.44 | 73.05 | 72.66 | 73.38 |
| | ACE05-Arg | 47.60 | 54.79 | 56.20 | **72.94** | 55.83 | 55.88 | 54.73 | 56.51 | 57.87 |
| | CASIE-Tgg | - | 69.33 | - | 67.80 | 71.73 | 71.81 | 71.60 | **73.09** | 71.40 |
| | CASIE-Arg | - | 61.30 | - | **63.53** | 63.26 | 61.27 | 61.04 | 60.44 | 58.87 |
| ABSA | 14-res | - | 74.52 | - | - | **77.26** | 75.06 | 74.24 | 76.05 | 75.89 |
| | 14-lap | - | 63.88 | - | - | **65.51** | 64.08 | 62.48 | 59.56 | 60.42 |
| | 15-res | - | 67.15 | - | - | **69.86** | 66.40 | 63.61 | 60.26 | 67.41 |
| | 16-res | - | 75.07 | - | - | **78.25** | 74.24 | 75.40 | 73.13 | 77.46 |
| Avg. | | - | 71.75 | - | - | 73.35 | 72.15 | 71.49 | 70.99 | 72.39 |

Table 3: Results on 13 IE benchmarks (ACE-Tgg and ACE-Arg are in the same dataset with different evaluation metrics). PT is the abbreviation of pretraining, and Inst. denotes the task instruction.

| | P | R | F1 |
|---|---|---|---|
| *Discontinuous NER: CADEC* | | | |
| BART-NER | 70.08 | 71.21 | 70.64 |
| W2NER | 74.09 | **72.35** | **73.21** |
| Mirrorw/ PT & Inst. | **74.83** | 65.45 | 69.83 |
| Mirrorw/o PT & Inst. | 68.80 | 68.38 | 68.59 |
| *N-ary Tuples: HyperRED* | | | |
| CubeRE | 66.39 | **67.12** | 66.75 |
| Mirrorw/ PT & Inst. | 71.29 | 62.46 | 66.58 |
| Mirrorw/o PT & Inst. | **75.41** | 61.14 | **67.53** |

Table 4: Results on multi-span and n-ary information extraction tasks. Tgg and Arg in Event Extraction refer to Trigger (Event Detection) and Argument (Event Argument Extraction), respectively.

| Task | Model | 1-shot | 5-shot | 10-shot | Avg. |
|---|---|---|---|---|---|
| NER CoNLL03 | UIE | 57.53 | 75.32 | 79.12 | 70.66 |
| | USM | 71.11 | **83.25** | 84.58 | 79.65 |
| | Mirror | **76.49** | 82.45 | **84.69** | **81.21** |
| RE CoNLL04 | UIE | 34.88 | 51.64 | 58.98 | 48.50 |
| | USM | **36.17** | **53.20** | **60.99** | **50.12** |
| | Mirror | 26.29 | 47.42 | 55.77 | 43.16 |
| Event Trigger ACE05 | UIE | 42.37 | 53.07 | 54.35 | 49.93 |
| | USM | 40.86 | 55.61 | 58.79 | 51.75 |
| | Mirror | **47.77** | **57.90** | **59.16** | **54.94** |
| Event Arg ACE05 | UIE | 14.56 | 31.20 | 35.19 | 26.98 |
| | USM | 19.01 | 36.69 | **42.48** | 32.73 |
| | Mirror | **23.18** | **37.74** | 39.20 | **33.38** |
| ABSA 16-res | UIE | 23.04 | 42.67 | 53.28 | 39.66 |
| | USM | 30.81 | **52.06** | 58.29 | 47.05 |
| | Mirror | **36.21** | 51.65 | **58.59** | **48.82** |

Table 5: Few-shot results on IE tasks. These datasets are not included in the pretraining phase of Mirror.

erage scores. Among all the four tasks, NER may be relatively easier for the model to deal with. The 10-shot NER score of Mirror is 84.69, while the fine-tuned Mirror on the full dataset gets an F1 score of 92.73. The gaps on other datasets between 10-shot and fully fine-tuned results are larger, indicating the task difficulties.

## 4.5 Zero-shot Results

Table 6 shows the zero-shot performances on 7 NER datasets. These datasets are not included in pretraining, and we use the pretrained Mirror to make predictions directly. The results show that Mirror outperforms USM by a large margin (an average F1 score of 9.44), and it is very competitive with InstructUIE (FlanT5-11B). Considering the model scale, Mirror is surprisingly good at zero-shot NER tasks. However, ChatGPT is very powerful in the zero-shot NER task and achieves absolute SOTA performance. Except for simple model scaling, we may need to collect a more diverse pretraining corpus for better results.

| Model | Movie | Restaurant | AI | Literature | Music | Politics | Science | Avg. |
|---|---|---|---|---|---|---|---|---|
| Davinci | 0.84 | 2.94 | 2.97 | 9.87 | 13.83 | 18.42 | 10.04 | 8.42 |
| ChatGPT | 41.00 | 37.76 | 54.40 | 54.07 | 61.24 | 59.12 | 63.00 | 52.94 |
| USM | 37.73 | 14.73 | 28.18 | **56.00** | 44.93 | 36.10 | 44.09 | 37.39 |
| InstructUIE | **63.00** | **20.99** | **49.00** | 47.21 | 53.61 | 48.15 | 49.30 | **47.32** |
| Mirror$_{direct}$ | 39.20 | 16.32 | 45.23 | 46.32 | **58.61** | **67.30** | **54.84** | 46.83 |

Table 6: Zero-shot results on 7 NER datasets. Results of Davinci and ChatGPT are derived from Wang et al. (2023). Mirror$_{direct}$ is the pretrained Mirror w/ Inst. while these datasets are not included in the pretraining phase.

| Model | SQuAD 2.0 (EM/F1) | CoLA (Mcc) | QQP (Acc) | MNLI (Acc) | SST-2 (Acc) | QNLI (Acc) | RTE (Acc) | MRPC (Acc) |
|---|---|---|---|---|---|---|---|---|
| BERT-large | 79.0 / 81.8 | 60.6 | 91.3 | - | 93.2 | 92.3 | 70.4 | 84.1 |
| RoBERTa-large | 86.5 / 89.4 | 68.0 | 92.2 | 90.2 | 96.4 | 93.9 | 86.6 | 88.8 |
| DeBERTa v3-large | 89.0 / 91.5 | 75.3 | 93.0 | 91.9 | 96.9 | 96.0 | 92.7 | 92.2 |
| Mirror$_{direct}$ | 40.4 / 67.4 | 63.9 | 84.8 | 85.9 | 93.6 | 91.6 | 85.9 | 89.2 |

Table 7: Results on MRC and classification tasks. We list Mirror performance on SQuAD 2.0 development set and GLUE development sets. Baseline results are derived from He et al. (2021). Because SQuAD v2 and GLUE datasets are included in Mirror pretraining for 3 epochs, we direct make inferences with the pretrained model (noted as Mirror$_{direct}$, the same model used in zero-shot NER), and do not perform further fine-tuning, while other baselines are fine-tuned with a full dataset on every single task.

## 4.6 Results on MRC and Classification

To show the model compatibility on extractive MRC and classification tasks, we conduct experiments on SQuAD v2 and GLUE language understanding benchmarks. The experimental results are demonstrated in Table 7. Comparing the results in He et al. (2021), we do not report performance on the STS-B dataset since Mirror's extraction paradigm does not support the regression task. Although Mirror$_{direct}$ does not perform full fine-tuning like the other systems, it still produces competitive results. It outperforms BERT-large on CoLA and SST-2 and is better than RoBERTa-large on MRPC. The results indicate that Mirror is capable of various tasks besides information extraction. We leave full fine-tuning for future work to improve Mirror performances.

## 4.7 Analysis on Label Span Types

Mirror adopts the leading token in schema labels (`[LC]`, `[LM]` and `[LR]`) as the label span that connects to target text spans. To analyze the effect of different label span types, we conduct experiments to change the leading token into a literal content string. In other words, in a NER task that extract *person* entities, we compare the effect of `[LM]` token and `person` string as the label span.

| w/ Inst. | Label Type | P | R | F1 |
|---|---|---|---|---|
| ✓ | Tag | 92.12 | 92.10 | 92.11 |
| | Content | 91.89 | 92.71 | 92.30 |
| ✗ | Tag | 92.79 | 93.15 | 92.97 |
| | Content | 91.91 | 92.58 | 92.25 |

Table 8: Results on different label span types. This experiment is conducted on the CoNLL03 dataset w/o pretraining.

The results are demonstrated in Table 8. We find that the label type does not bring too many differences. In Mirror w/ Inst., the literal content string is slightly better than bare tags with only a 0.19 F1 score advantage. While in Mirror w/o Inst., the tag-based method surpasses the content-based method by 0.72 F1 scores. Similar to Baldini Soares et al. (2019), these results show that although the label tag is a simple token without pretraining, it does not affect the model's ability to incorporate features from global and local contexts.

## 4.8 Analysis on Pretraining Datasets

Traditionally, the classification task is different from the extraction task as they optimize different objectives. Since Mirror unifies the two tasks

| Pretrain Data | NER (F1) CoNLL03 | RE (F1) NYT | MRC (F1) SQuAD v2 | Cls (Acc) MRPC | Average |
|---|---|---|---|---|---|
| All | 66.91 | 69.67 | 67.39 | 89.22 | 73.30 |
| w/o Cls | 66.82 | 57.03 | 67.14 | 0.00 | 47.75 |
| w/o IE | 0.00 | 0.00 | 68.77 | 89.22 | 39.50 |
| w/o Span | 66.76 | 54.76 | 0.00 | 87.50 | 52.26 |

Table 9: Ablation study on the pretraining data. We evaluate the pretrained Mirror$_{\text{direct}}$ without further fine-tuning.

into one framework, it is interesting to find how they affect each other in the pretraining phase. We provide an ablation study on different types of pretraining data in Table 9. It is surprising that pretraining on classification datasets help improve the extraction tasks, and relation extraction is the most affected one. This may be due to the similarity between relation labels and semantic class labels. It is also interesting that span-based datasets (e.g. MRC datasets) are beneficial to the classification task (87.50 → 89.22). Overall, all kinds of the pretraining datasets bring greater mutual benefits and improve the model performance.

### 4.9 Analysis on Inference Speed

We conduct speed tests on the CoNLL03's validation set with one NVIDIA V100 GPU under the same environment. The results are presented in Table 10. Compared to the popular generative T5-large UIE model (Lu et al., 2022), our model is up to 32.61 times faster when inference, and the advantage grows when increasing the batch size from 1 to 2.

| batch size | UIE | Mirror | Speed-Up |
|---|---|---|---|
| 1 | 0.21 | 5.68 | 27.24 |
| 2 | 0.32 | 10.56 | 32.61 |

Table 10: Inference speed (instances per second) test on CoNLL03 validation set.

## 5 Conclusion

We propose Mirror, a schema-guided framework for universal information extraction. Mirror transforms IE tasks into a unified multi-slot tuple extraction problem and introduces the multi-span cyclic graph to represent such structures. Due to the flexible design, Mirror is capable of multi-span and n-ary extraction tasks. Compared to previous systems, Mirror supports not only complex information extraction but also MRC and classification tasks. We manually collect 57 datasets for pretraining and conduct experiments on 30 datasets across 8 tasks. The experimental results show good compatibility, and Mirror achieves competitive performances with state-of-the-art systems.

## Limitations

Content input length: Due to the backbone DeBERTa model constraint, the maximal sequence length is 512 and can hardly extend to longer texts. This limits the exploration of tasks with many schema labels and document-level IE.

Multi-turn result modification: Mirror predicts the multi-span cyclic graph in a paralleled non-autoregressive style. Although it is efficient in training and inference, it may lack global history knowledge from previous answers.

Data format unification: There are many IE tasks, and the formats may vary a lot. Although the current unified data interface supports most common tasks, it may not be practical for some tasks.

Lack of large-scale event datasets for pretraining: There are many NER and RE datasets. However, there are few large-scale event extraction corpus with high diversity in domains and schemas, which may limit the model performance on event-relevant information extraction tasks.

## Acknowledgments

This work is supported by the National Natural Science Foundation of China (Grant No. 61936010) and Provincial Key Laboratory for Computer Information Processing Technology, Soochow University. This work is also supported by Collaborative Innovation Center of Novel Software Technology and Industrialization, the Priority Academic Program Development of Jiangsu Higher Education Institutions, and the joint research project of Huawei Cloud and Soochow University. We would also like to thank the anonymous reviewers for their insightful and valuable comments.

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

## A  Comparisons on Information Indexing Strategies

UIE (Lu et al., 2022) provides the extracted information's positions based on string matching. However, this strategy is not accurate and contains ambiguities. To investigate the matching accuracy, we take the NER task as an example and use golden entity strings to calculate the upper bound F1 scores of different UIE string matching strategies. The table below shows that the upper bounds are quite low on the datasets (<30%). This indicates that obtaining positions via string matching is ineffective and has serious ambiguity problems.

| NER | ACE04 | ACE05 | CoNLL03 |
|---|---|---|---|
| Mirror | 100.00 | 100.00 | 100.00 |
| UIE-first | 13.31 | 14.51 | 27.67 |
| UIE-longer-first | 14.55 | 16.21 | 27.97 |

Table 11: Upper bound of different string matching strategies on NER.

TANL (Paolini et al., 2021) can provides exact positions in NER since it generates the enclosure tags. However, it still faces the ambiguity problem when two entities have the same string in joint entity relation extraction because the tail entity is a generated text corresponding to an enclosed head entity (refer to section 3 in the TANL paper). We also calculate the upper bound F1 scores of relation extraction in a TANL manner, and the results show it does not ideally generate perfect positions.

| RE | ACE05 | CoNLL04 | NYT | SciERC |
|---|---|---|---|---|
| Mirror | 100.00 | 100.00 | 100.00 | 100.00 |
| TANL | 96.91 | 96.95 | 100.00 | 99.64 |

Table 12: Upper bound of relation extraction with Mirror and TANL position indexing strategies.

## B  Hyper-parameter Settings

Table 13 shows the hyper-parameters in our experiments. For few-shot experiments, we follow Lu et al. (2022) and generate 1-, 5-, 10-shot data with 5 seeds.

| Item | Setting |
|---|---|
| warmup proportion | 0.1 |
| pretraining epochs | 3 |
| fine-tuning epochs | 20 |
| fine-tuning epoch patience | 3 |
| few-shot epochs | 200 |
| few-shot epoch patience | 10 |
| batch size | 8 |
| PLM learning rate | 2e-5 |
| PLM weight decay | 0.1 |
| others learning rate | 1e-4 |
| max gradient norm | 1.0 |
| $d_h$ | 1024 |
| $d_b$ | 512 |
| dropout | 0.3 |

Table 13: Hyper-parameter settings.

## C  Dataset Statistics

This section contains detailed statistics for pretraining datasets and fine-tuning datasets. Pretraining data statistics are listed in Table 14, 16, 17, 18 and 15. For the sampling number $N_{\max}$ of each kind of dataset, please refer to Table 2. When collecting pretraining data, we refer to the datasets mentioned in Therasa and Mathivanan (2022) and Yang et al. (2022). Downstream data statistics are listed in Table 19. We also provide direct inference results with the pretrained Mirror model in Table 19.

## D  Case Study

We provide some interesting cases across different tasks with the pretrained Mirror w/ Inst. to manually evaluate its versatility on various tasks under zero-shot settings. The model inputs & outputs are presented in Table 20.

| Name | #Instruction | #Instance |
|---|---|---|
| ag_news | 5 | 5,000 |
| ANLI♣ | 29 | 15,000 |
| ARC | 3,361 | 3,370 |
| CoLA | 43 | 5,000 |
| CosmosQA | 4,483 | 5,000 |
| cos_e | 5,000 | 5,000 |
| dbpedia | 6 | 5,000 |
| DREAM | 3,842 | 5,000 |
| hellaswag | 20 | 5,000 |
| IMDB | 26 | 5,000 |
| MedQA | 5,000 | 5,000 |
| MNLI | 29 | 5,000 |
| MRPC | 40 | 3,668 |
| MultiRC | 4,999 | 5,000 |
| OpenBookQA | 4,835 | 4,957 |
| QASC | 4,832 | 5,000 |
| QNLI | 31 | 5,000 |
| QQP | 40 | 5,000 |
| RACE | 4,482 | 5,000 |
| RACE-C | 4,782 | 5,000 |
| ReClor | 3,368 | 4,638 |
| RTE | 29 | 2,490 |
| SciQ | 4,989 | 5,000 |
| SNLI | 29 | 5,000 |
| SST-2 | 26 | 5,000 |
| Winogrande | 20 | 5,000 |
| WNLI | 31 | 635 |
| Total | 54,070 | 134,758 |

Table 14: Pretraining data statistics on classification. The maximal sampling number $N_{max}$ for each dataset is 5,000. ♣: ANLI contains 3 subsets, so the total number is greater than 5,000.

| Name | #Instruction | #Instance |
|---|---|---|
| PHEE | 40 | 2,898 |
| Total | 40 | 2,898 |

Table 15: Pretraining data statistics on EE. Due to the scarcity of EE datasets, we sample all the instances ($N_{max} = \infty$).

| Name | #Instruction | #Instance |
|---|---|---|
| AnatEM | 42 | 5,861 |
| bc2gm | 42 | 12,500 |
| bc4chemd | 42 | 20,000 |
| bc5cdr | 42 | 4,560 |
| Broad_Tweet_Corpus | 42 | 5,334 |
| FabNER | 42 | 9,435 |
| FindVehicle | 42 | 20,000 |
| GENIA | 42 | 15,023 |
| HarveyNER | 42 | 3,967 |
| MultiNERD | 42 | 20,000 |
| NCBIdiease | 42 | 5,432 |
| ontoNotes5 | 42 | 20,000 |
| TweetNER7 | 42 | 7,103 |
| WikiANN_en | 42 | 20,000 |
| WNUT-16 | 42 | 2,394 |
| Total | 42 | 171,609 |

Table 16: Pretraining data statistics on NER. The maximal sampling number $N_{max}$ for each dataset is 20,000.

| Name | #Instruction | #Instance |
|---|---|---|
| ADE_corpus | 9 | 3,417 |
| FewRel | 9 | 20,000 |
| GIDS | 9 | 8,526 |
| kbp37 | 9 | 15,807 |
| New-York-Times-RE | 9 | 20,000 |
| NYT11HRL | 9 | 20,000 |
| semeval | 9 | 8,000 |
| WebNLG | 9 | 5,019 |
| Wiki-ZSL♣ | 9 | 23,107 |
| Total | 9 | 123,876 |

Table 17: Pretraining data statistics on RE. The maximal sampling number $N_{max}$ for each dataset is 20,000.

| Name | #Instruction | #Instance |
|---|---|---|
| BiPaR | 11,524 | 11,668 |
| ms_marco_v2.1 | 20,000 | 20,000 |
| newsqa | 19,659 | 20,000 |
| squad_v2 | 19,998 | 20,000 |
| SubjQA | 4,060 | 13,990 |
| Total | 75,220 | 85,658 |

Table 18: Pretraining data statistics on MRC. The maximal sampling number $N_{max}$ for each dataset is 20,000.

| Task | Dataset | Citation | Metric | #Train | #Dev | #Test | Included in PT | Mirror$_{\text{direct}}$ |
|---|---|---|---|---|---|---|---|---|
| | ACE04 | Mitchell et al. (2005) | Entity Micro F1 | 6,202 | 745 | 812 | ✗ | 21.49 |
| NER | ACE05 | Walker et al. (2006) | Entity Micro F1 | 7,299 | 971 | 1,060 | ✗ | 18.70 |
| | CoNLL03 | Tjong Kim Sang and De Meulder (2003) | Entity Micro F1 | 14,041 | 3,250 | 3,453 | ✗ | 66.91 |
| | ACE05 | Walker et al. (2006) | Triplet Micro F1 | 10,051 | 2,420 | 2,050 | ✗ | 0.51 |
| RE | CoNLL04 | Roth and Yih (2004) | Triplet Micro F1 | 922 | 231 | 288 | ✗ | 1.40 |
| | NYT | Riedel et al. (2010) | Triplet Micro F1 | 56,196 | 5,000 | 5,000 | ✗ | 69.67 |
| | SciERC | Luan et al. (2018) | Triplet Micro F1 | 1,861 | 275 | 551 | ✗ | 0.00 |
| EE | ACE05 | Walker et al. (2006) | Trigger & Argument Micro F1 | 19,216 | 901 | 676 | ✗ | 3.99/0.00 |
| | CASIE | Satyapanich et al. (2020) | Trigger & Argument Micro F1 | 11,189 | 1,778 | 3,208 | ✗ | 2.13/0.00 |
| | 14-res | Pontiki et al. (2014) | Triplet Micro F1 | 1,266 | 310 | 492 | ✗ | 0.00 |
| ABSA | 14-lap | Pontiki et al. (2014) | Triplet Micro F1 | 906 | 219 | 328 | ✗ | 0.00 |
| | 15-res | Pontiki et al. (2015) | Triplet Micro F1 | 605 | 148 | 322 | ✗ | 0.00 |
| | 16-res | Pontiki et al. (2016) | Triplet Micro F1 | 857 | 210 | 326 | ✗ | 0.00 |
| Discontinuous NER | CADEC | Karimi et al. (2015) | Entity Micro F1 | 5,340 | 1,097 | 1,160 | ✗ | 52.34 |
| Hyper RE | HyperRED | Chia et al. (2022) | Tuple Micro F1 | 39,840 | 4,000 | 1,000 | ✗ | 0.00 |
| | Movie | Liu et al. (2013) | Entity Micro F1 | 9,774 | 2,442 | 2,442 | ✗ | 39.24 |
| | Restaurant | Liu et al. (2013) | Entity Micro F1 | 7,659 | 1,520 | 1,520 | ✗ | 16.17 |
| | AI | Liu et al. (2021) | Entity Micro F1 | 100 | 350 | 431 | ✗ | 45.91 |
| Zero-shot NER | Literature | Liu et al. (2021) | Entity Micro F1 | 100 | 400 | 416 | ✗ | 46.77 |
| | Music | Liu et al. (2021) | Entity Micro F1 | 100 | 380 | 465 | ✗ | 59.12 |
| | Politics | Liu et al. (2021) | Entity Micro F1 | 199 | 540 | 650 | ✗ | 67.27 |
| | Science | Liu et al. (2021) | Entity Micro F1 | 200 | 450 | 543 | ✗ | 54.42 |
| MRC | SQuAD v2.0 | Rajpurkar et al. (2018) | Exact Match & F1 | 86,821 | 5,928 | - | ✓ | 40.35/67.39 |
| | CoLA | Warstadt et al. (2019) | Matthew's Correlation Coefficient | 8,551 | 527 | - | ✓ | 63.91 |
| | QQP | Wang et al. (2019) | Accuracy | 363,846 | 40,430 | - | ✓ | 84.84 |
| | MNLI | Williams et al. (2018) | Accuracy | 392,702 | 9,815 | - | ✓ | 85.90 |
| Classification | SST-2 | Socher et al. (2013) | Accuracy | 67,350 | 873 | - | ✓ | 93.58 |
| | QNLI | Wang et al. (2019) | Accuracy | 104,743 | 5,463 | - | ✓ | 91.62 |
| | RTE | Wang et al. (2019) | Accuracy | 2,490 | 277 | - | ✓ | 85.92 |
| | MRPC | Dolan and Brockett (2005) | Accuracy | 3,668 | 408 | 1,725 | ✓ | 89.22 |

Table 19: Data statistics on downstream tasks. Included in PT stands for whether the dataset is included in the data pretraining corpus. Mirror$_{\text{direct}}$ is the model trained on the pretraining corpus.

---

*Classification (Multi-choice MRC)*

**Input**  [I] Mirror Mirror on the wall, who's the fairest of them all? [LC] Evil Queen [LC] Snow White

**Output**  [LC]$_{\text{Snow White}}$

---

*Extractive MRC*

**Input**  [I] Mirror Mirror on the wall, who's the fairest of them all? [TP] Evil Queen is jealous of Snow White's beauty.

**Output**  Snow White

---

*Named Entity Extraction*

**Input**  [I] Mirror Mirror, please help me extract all the model names. [LM] model name [TL] LLaMA and OPT are open-sourced large language models.

**Output**  [LM]$_{\text{LLaMA}}$, [LM]$_{\text{OPT}}$

---

*Relation Extraction*

**Input**  [I] Mirror Mirror, please help me extract the entity relationship triplet. [LR] break up [TL] The drama surrounding the high-profile divorce between Hollywood actors Johnny Depp and Amber Heard appears to be over as the couple reportedly reached an amicable settlement.

**Output**  ([LR]$_{\text{break up}}$, Amber Heard, Johnny Depp)

---

Table 20: Case results obtained by the pretrained Mirror w/ Inst. The name of our proposed Mirror is borrowed from the magic mirror in *Snow White and the Seven Dwarfs*. We hope to build a universal model that can help more people solve more problems.

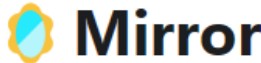

**Mirror**

🪞 Mirror can help you deal with a wide range of Natural Language Understanding and Information Extraction tasks.

**Instruction**

Mirror Mirror, please help me extract the entity relationship triplet.

**Schema Labels**

Split with `#` for multiple inputs

For entities, relations or classification, input `{"ent|rel|cls": ["cls1", "type2"]}` .

For events and hyper relations, input `{"type": ["role1", "role2"]}` .

{"rel": ["break up"]}

**Text**

The drama surrounding the high-profile divorce between Hollywood actors Johnny Depp and Amber Hea

**Background**

[ Reset ]  [ Clear Output ]  [ **Ask Mirror** ]

**Output**

| Item | Predicted |
|------|-----------|
| rel | [ [ "break up", [ "Amber Heard", [ 33, 34 ] ], [ "Johnny Depp", [ 30, 31 ] ] ] ] |

Made by Mirror Team w/ 💖

Figure 4: Mirror toolkit demonstration. The predicted relation label is converted to label string *break up*. The positions shown in the predicted results are counted by tokens, so they do not match the input string characters. You can find the pretrained model weights and demo code in the repository and deploy it on your own machine: https://github.com/Spico197/Mirror