# OpenReview forum: "Mirror: A Universal Framework for Various Information Extraction Tasks"
_EMNLP/2023/Conference — EMNLP 2023 Main_

### Official Review · Reviewer_59YZ · 2023-08-03

**Typos Grammar Style And Presentation Improvements:** 1.	“Extractive Pretrained Language Mo…
**Soundness:** 4

**Excitement:**

2: Mediocre: This paper makes marginal contributions (vs non-contemporaneous work), so I would rather not see it in the conference.

**Paper Topic And Main Contributions:**

This paper presents a novel framework called Mirror for information extraction (IE) tasks, which reorganizes IE problems as a multi-span cyclic graph extraction problem and devise a non-autoregressive graph decoding algorithm to extract all spans in a single step.
This graph structure is incredibly versatile, supporting not only complex IE tasks but also machine reading comprehension and classification tasks. Furthermore, the authors have manually constructed a pre-training corpus containing 57 datasets and have conducted experiments on 30 datasets across 8 downstream tasks, demonstrating the model's compatibility and outperforming or reaching competitive performance under the zero/few-shot settings. This paper also provides a complete code and dataset as well as an interactive interface for future work. This is beneficial for future research.


**Questions For The Authors:**

1.	Why treat the MRC task's question as an instruction, rather than a label schema?  USM/MRC-EE treat the MRC task's question as a special type of entity/event, and their experiments have proven that this can enhance the entity/event argument model's ability to handle zero-shot scenarios by leveraging the diversity of the question. However, Mirror's approach does not benefit in this way.
https://aclanthology.org/2020.emnlp-main.128.pdf
2.	What distinguishes Extractive Question Answering (QA) from extractive Machine Reading Comprehension (MRC) tasks, and why is this distinction repeatedly stressed in the paper? Within the framework of Mirror, they represent the same task: given a query, the goal is to predict the boundaries of the answer and extract the answer.


**Reasons To Accept:**

1.	This paper proposes a framework that is suitable for extraction, MRC, and classification tasks and further introduces an Instruction-enhanced model building on the traditional BERT-Style framework.
2.	The paper conducts detailed experiments on massive datasets and has strong few-shot and zero-shot performance. Moreover, it introduces discontinuous entity extraction for the first time in universal information extraction tasks.


**Reasons To Reject:**

The comparisons made in Table 1 lack rigor. In Table 1, the authors purport that Mirror, in contrast to other universal information extraction models and structure prediction models, can be applied to not only complex IE tasks but also MRC and classification tasks. Yet, in theory, these tasks can also be accomplished by other frameworks.
Firstly, all of these frameworks can perform MRC tasks as they are all capable of extracting a single text span given a query. Furthermore, frameworks such as TANL and DeepStruct, despite being structure prediction models, are not inherently incapable of performing classification tasks. TANL and DeepStruct have been tested on relation classification tasks, and by removing entity identifiers, they both can perform text classification, similar to the original T5 model.

The authors also contend that a significant issue with generation-style methods is their failure to accurately predict the location of structured information, which is critical for NER tasks and fair evaluations. However, both UIE (Lu et al., 2022) and TANL (Paolini et al., 2021) utilize Token Position as the extraction result. UIE employs the string match method to recover the span's position, while TANL generates identifiers encompassed by specific tokens, thereby effectively obtaining location information.

Although Mirror serves as a universal framework that can simultaneously handle classification and extraction, I did not see any impact of pre-training with classification data on model performance. Moreover, although classification and extraction are similar in form, they do not focus on the same information. Classification tasks pay more attention to the overall information of the text, while extraction tasks focus more on specific local information. It is recommended that the authors conduct ablation experiments on pre-training data according to different task types to clarify the role of each pre-training task part.


**Reproducibility:**

4: Could mostly reproduce the results, but there may be some variation because of sample variance or minor variations in their interpretation of the protocol or method.

**Reviewer Confidence:**

5: Positive that my evaluation is correct. I read the paper very carefully and I am very familiar with related work.

---

> ### Author Rebuttal · Authors · 2023-08-29
>
> Thank you for the helpful suggestions and elaborate comments! In the following, your questions are first stated and then followed by our responses.
>
> **Q: The comparisons made in Table 1 lack rigor. In theory, MRC and classification tasks can also be accomplished by other frameworks.**
>
> It is a valid point that the MRC and classification tasks could be theoretically accomplished by multiple frameworks, such as extraction models and flexible generation-based models. However, in the actual usage of these methods, there are many potential problems.
> - For the MRC task: It is slightly different from NER as there is a query per instance without span types. For triple-based universal IE frameworks (e.g., UIE, DeepStruct), the query may be integrated into the input texts directly, while the type-free span is difficult to be represented as a triple. USM is pretrained on MRC datasets, but the results are absent from their paper. So, we tag it as "theoretically supports MRC" in Table 1.
> - For DeepStruct and TANL frameworks on the classification task: The task is not officially supported as no relevant experiments are conducted in their papers. As you have said, there are some potential difficulties in supporting the task since we have to set up new input and output formats and devise new corresponding structure decoding strategies. We will revise Table 1 with a more fine-grained comparison to clarify the distinctions and commonalities between the proposed Mirror framework and other approaches.
>
> **Q: Information positions could be obtained from generation-style methods.**
>
> Thank you for pointing out the position information and its utilization in UIE and TANL. There is no such issue in Mirror and USM, while it is a common problem in generation-based methods.
> - For string matching in UIE, it is not perfect and may bring ambiguities. To investigate the matching accuracy, we take the NER task as an example and use golden entity strings to calculate the upper bound F1 scores of different UIE string matching strategies. The table below shows that the upper bounds are quite low on the datasets (<30%). This indicates that obtaining positions via string matching is ineffective and has serious ambiguity problems.
> | NER           | ACE04 ↑ | ACE05 ↑ | CoNLL03 ↑ |
> |------------------|--------:|--------:|----------:|
> | Mirror           |  100.00 |  100.00 |    100.00 |
> | UIE-first        |   13.31 |   14.51 |     27.67 |
> | UIE-longer first |   14.55 |   16.21 |     27.97 |
>
> - We acknowledge that TANL can provide some positions since it uses the enclosure tags. However, it still faces the ambiguity problem when two entities have the same string in joint entity relation extraction because the tail entity is a generated text corresponding to an enclosed head entity (refer to section 3 in the TANL paper). We also calculate the upper bound F1 scores of relation extraction in a TANL manner, and the results show it does not ideally generate accurate positions.
> | RE | ACE05 ↑ | CoNLL04 ↑ | NYT ↑  | SciERC ↑ |
> |--------|--------:|----------:|-------:|---------:|
> | Mirror |  100.00 |    100.00 | 100.00 |   100.00 |
> | TANL   |   96.91 |     96.95 |    100.00 |    99.64 |
>
> - About the crossmarks in Table 1, our original intention was to compare unambiguous position generation methods. We will carefully revise our comparison to ensure that the statement accurately reflects the differences in how position information is provided across different frameworks instead of simple checkmarks and crossmarks.
>
> **Q:  It is recommended that the authors conduct ablation experiments on pre-training data according to different task types to clarify the role of each pre-training task part.**
>
> Thanks for the suggestion! We conduct ablation experiments on the pretraining data to better understand how different pre-training tasks contribute to the performance. As the table below shows, both classification and MRC data improve the IE performances, and we think Mirror could benefit from the classification task to gather more global features. We will conduct more ablation studies and add these results and corresponding analyses to the revised version. We believe that this will enhance the transparency of our approach and its effectiveness in handling diverse tasks.
>
> |               | NER - CoNLL03 (F1) | Rel - NYT (F1) | MRC - SQuAD v2 (F1) | Cls - MRPC (Acc) | Average |
> |---------------|-------------------:|---------------:|--------------------:|-----------------:|--------:|
> | Mirror-direct |              66.91 |          69.67 |               67.39 |            89.22 |   73.30 |
> | w/o Cls       |              66.82 |          57.03 |               67.14 |             0.00 |   47.75 |
> | w/o IE        |               0.00 |           0.00 |               68.77 |            89.22 |   39.50 |
> | w/o Span      |              66.76 |          54.76 |                0.00 |            87.50 |   52.26 |
>
> **Q: Why treat the MRC task's question as an instruction, rather than a label schema?**
>
> Thank you for the question and the reference! We want to ensure every task is instructed, so the instruction section is different from label schemas. In addition, MRC's natural language questions are more similar to instructions rather than discrete labels, so we differentiate those questions from labels when constructing inputs to the model.
>
> **Q: What distinguishes extractive Question Answering (QA) from extractive Machine Reading Comprehension (MRC) tasks, and why is this distinction repeatedly stressed in the paper?**
>
> Sorry for the confusion. We intended to emphasize they are the same task. We will address the problem in the revised version with more caution.
>
> **Q: The writing needs to be improved.**
>
> We appreciate your feedback on the writing style and grammatical errors. We will thoroughly review and improve the writing to ensure the paper is presented in a clear and polished manner.

---

### Official Review · Reviewer_aSvu · 2023-08-05

**Soundness:** 4

**Excitement:**

3: Ambivalent: It has merits (e.g., it reports state-of-the-art results, the idea is nice), but there are key weaknesses (e.g., it describes incremental work), and it can significantly benefit from another round of revision. However, I won't object to accepting it if my co-reviewers champion it.

**Paper Topic And Main Contributions:**

The paper proposes, Mirror, a framework that can handle complex multi-span extraction, n-ary extraction, machine reading comprehension (MRC). It formulates IE tasks as a unified multi-slot tuple extraction problem and transform those tuples into multi-span cyclic graphs. This graph structure is rather flexible and scalable. It can be applied to not only complex IE tasks but also MRC tasks. Extensive experiments on 30 datasets from 8 tasks, and the results show that our model achieves competitive results under few-shot and zero-shot settings.

**Reasons To Accept:**

1. well-designed graph-based IE decoding method.
2. well-formed layout.
3. The motivation is clear and persuasive in a certain level.

**Reasons To Reject:**

1. The performance is not strong enough
2. I think the proposed methd significantly underperforms USM (Table 3)
3. Although the paper claims that the proposed method can conduct MRC and classification tasks, it underperforms direct fine-tuning, so I think this claim is trivial.



**Reproducibility:**

4: Could mostly reproduce the results, but there may be some variation because of sample variance or minor variations in their interpretation of the protocol or method.

**Reviewer Confidence:**

4: Quite sure. I tried to check the important points carefully. It's unlikely, though conceivable, that I missed something that should affect my ratings.

---

> ### Author Rebuttal · Authors · 2023-08-29
>
> Thank you for the feedback! In the following, your questions are first stated and then followed by our responses.
>
> **Q: The performance is not strong enough. I think the proposed method significantly underperforms USM (Table 3)**
> - Mirror presents competitive finetuing performance with USM. As shown in Table 3, Mirror outperforms USM on five datasets (i.e., NER-ACE04, RE-ACE05, RE-NYT, RE-SciERC, EE-CASIE-Trigger), and the average result for Mirror is only 0.96%, slightly lower than USM.
> - Mirror significantly surpasses USM on few-shot and zero-shot settings. As presented in Table 5, Mirror outperforms USM on few-shot NER, Event Extraction, and ABSA tasks, and surpasses USM with 9.44% F1 on the zero-shot NER, as illustrated in Table 6.
> - Mirror supports more tasks than USM, such as discontinuous NER, n-ary RE, MRC, and classification.
>
> **Q: Although the paper claims that the proposed method can conduct MRC and classification tasks, it underperforms direct fine-tuning, so I think this claim is trivial.**
>
> Since part of the MRC and classification datasets are included in the pre-training phase, we only report Mirror-direct in Table 7 without fully fine-tuning. After we fully fine-tuning Mirror for 5 epochs (Mirror-FT), we found the performance similar to RoBERTa-large. We will conduct more experiments on MRC and classification tasks in the revised version, addressing your concerns more comprehensively.
>
> |               | CoLA (Mcc) | QQP (Acc) | QNLI (Acc) | RTE (Acc) |
> |---------------|-----------:|----------:|-----------:|----------:|
> | RoBERTa-large |      68.0  |     92.2  |      93.9  |     86.6  |
> | Mirror-direct |      63.9  |     84.8  |      91.6  |     85.9  |
> | Mirror-FT     |      70.2  |     91.5  |      93.6  |     86.6  |

---

### Official Review · Reviewer_z1TA · 2023-08-05

**Soundness:** 4

**Excitement:**

4: Strong: This paper deepens the understanding of some phenomenon or lowers the barriers to an existing research direction.

**Paper Topic And Main Contributions:**

This paper introduces a new universal information extraction (UIE) model called Mirror. Unlike autoregressive UIE, this paper proposes a paradigm that treats IE problems as multi-span cyclic graphs extraction. The model is evaluated on a wide variety of task including joint IE, entity extraction, classification and MRC.

**Questions For The Authors:**

* The authors acknowledge certain limitations of the model, particularly its scalability concerns with respect to handling long text or document-level inputs due to the transformer's context size limit. The authors should also point out that the model would not scale well in a setup where the number entity types and relations types are very large (especially for end-to-end RE).
* Figure 3 caption should be more detailed, where each step is clearly described
* The complexity of the decoding algorithm (ALG 1) should be clearly stated. What is the speed-up of this model compared to autoregressive UIE models ?


**Reasons To Accept:**

* The idea is interesting, the results are strong, and the experiments are well executed with thorough analysis.
* It is nice to see that instruction tuning is applied to bidirectional transformers, allowing prediction in a single shot in contrast to autoregressive decoding. The model also obtain competitive result under few-shot and zero-shot setting.

**Reasons To Reject:**

* Due to the pertaining on multiple labelled datasets, the comparaison with other UIE approaches is not really fair. I expect that other UIE baselines would perform similarly (or better) if pertained using the instruction same dataset.


**Reproducibility:**

3: Could reproduce the results with some difficulty. The settings of parameters are underspecified or subjectively determined; the training/evaluation data are not widely available.

**Reviewer Confidence:**

3: Pretty sure, but there's a chance I missed something. Although I have a good feel for this area in general, I did not carefully check the paper's details, e.g., the math, experimental design, or novelty.

---

> ### Author Rebuttal · Authors · 2023-08-29
>
> Thank you for the thorough comments and constructive suggestions! In the following, your questions are first stated and then followed by our responses.
>
> **Q: Due to the pertaining on multiple labelled datasets, the comparison with other UIE approaches is not really fair.**
>
> It is difficult to pretrain other baseline models with our dataset since they are not designed to cover all the tasks, especially the discontinuous NER, n-ary RE, MRC, and classification tasks. Furthermore, as other models' pretraining datasets are not fully open-sourced, it is hard to pretrain Mirror with those data. Compared to other pretraining datasets, as shown in the table below, Mirror applies less pretraining data while supporting more tasks. While we acknowledge that there might be variations in performance based on pretraining data, we aimed to highlight the versatility and effectiveness of our proposed Mirror framework across a wide range of tasks. We will release our datasets to help readers reproduce our results.
>
> | Methods                | Pre-trainig tasks                                                                                                      | #Total Instance | #NER Instance | #RE Instance | #MRC Instance |
> |------------------------|------------------------------------------------------------------------------------------------------------------------|----------------:|--------------:|-------------:|--------------:|
> | UIE (Lu et al., 2022)  | Distantly-supervised NER and RE; ConceptNet and Wikidata for structure decoding; Causal language modeling on Wikipedia |         Unknown |       Unknown |      Unknown |          None |
> | USM (Lou et al., 2023) | NER, RE, MRC                                                                                                           |            611k |           60k |         356k |          195k |
> | Mirror (ours)          | NER, RE, EE, Cls, MRC                                                                                                  |            519k |          172k |         124k |           86k |
>
> **Q: The authors should also point out that the model would not scale well in a setup where the number of entity types and relations types are very large (especially for end-to-end RE).**
>
> Thank you for the suggestion and emphasizing the context limitation concerns we mentioned in the paper. Actually, all schema-guided end-to-end IE systems suffer from context limitation when scaling the number of labels. We agree that the scalability limitations are important to consider in future work, especially in scenarios involving long texts or document-level inputs, as well as tasks with a large number of entity and relation types. We will explicitly mention these limitations in the revised version.
>
> **Q: Figure 3 caption should be more detailed, where each step is clearly described.**
>
> Thanks for your suggestion. We will add the following explanation in the revised version: "Mirror first constructs inputs for each task, then utilizes a pretrained language model to predict the adjacency matrix via the biaffine attention. After that, final results are decoded from the adjacency matrix directly."
>
> **Q: The complexity of the decoding algorithm (ALG 1) should be clearly stated. What is the speed-up of this model compared to autoregressive UIE models?**
>
> Compared to the popular UIE model (Lu et al., 2022), our model is up to 32.61 times faster when inference, and the advantage still holds when the batch size increases. The speed tests are conducted with one NVIDIA V100 under the same hardware and software environment.
>
> | batch size | UIE (instances/second) | Mirror (instances/second) | Speed-up |
> |-----------:|-----------------------:|--------------------------:|---------:|
> |          1 |                   0.21 |                      5.68 |    27.24 |
> |          2 |                   0.32 |                     10.56 |    32.61 |
>
> **Q: Could reproduce the results with some difficulty.**
>
> Our hyper-parameter settings are available in section 4.1 and Appendix A. Meanwhile, all our pretraining datasets are publicly available, and the statistics are provided in Appendix B. We will open-source all the code and pretraining datasets for easily reproducing our results.

---

### Meta-Review · Area_Chair_De4T · 2023-09-16

**Recommendation:** 5

**Metareview:**

All reviewers agree that this work has made some interesting contributions to the field of IE by introducing the unified approach that casts various IE problems into a span graph prediction problem. The authors used a non-auto-regressive approach, which to me is the right way to go. There are useful suggestions for strengthening the work with additional experiments, which I suggest the authors incorporate into the final version.

---

### Decision · Program_Chairs · 2023-10-07

**Decision:**

Accept-Main

**Comment:**

All reviewers agree that this work has made some interesting contributions to the field of IE by introducing the unified approach that casts various IE problems into a span graph prediction problem. The authors used a non-auto-regressive approach, which to me is the right way to go. There are useful suggestions for strengthening the work with additional experiments, which I suggest the authors incorporate into the final version.